# Populations of arable weed species show intra-specific variability in germination base temperature but not in early growth rate

**Jana Bürger** [1]*, **Andrey V. Malyshev**[2], **Nathalie Colbach**[3]

**1** Crop Health, Faculty of Agricultural and Environmental Sciences, University of Rostock, Rostock, Germany, **2** Experimental Plant Ecology, Institute of Botany and Landscape Ecology, University of Greifswald, Greifswald, Germany, **3** Agroécologie, AgroSup Dijon, INRAE, Univ. Borgogne, Univ. Bourgogne Franche-Comté, Dijon, France

* jana.buerger@uni-rostock.de

## Abstract

Key plant traits affecting growth performance can differ among and within species, influencing competitive plant community dynamics. We determined the intra-specific variability of germination base temperature among 13 arable weed species and the seedlings' early post-emergence relative growth rate among 21 species in climate chamber and green house experiments. Intra-specific variability was quantified with two seed populations (originating from contrasting climate in Germany & France) for the germination base temperature of 6 species and for the early growth rate of 16 species. Inter-specific variability for both traits was always higher than intra-specific variability. Within a given species, we found that germination base temperatures were higher in seeds stemming from colder climate populations. Seedling relative growth rates did not differ between seed populations. Models simulating weed growth should reflect these differences in germination traits among populations, especially when they are used for weed community assembly studies in a local to regional extent.

**Data Availability Statement:** All data has been uploaded to the DRYAD repository. doi: 10.5061/dryad.nzs7h44p0.

**Funding:** JB was funded by Deutsche Forschungsgemeinschaft (https://www.dfg.de/)

## Introduction

Over the last decades, there has been an increase in trait-based analyses and modelling, both in ecology in general [1] and in weed ecology in particular [2, 3] followed by a debate on the importance and implications of intra-specific trait variability. Initially, the assumption of many trait-based approaches was that intra-specific variability is negligible compared to the inter-specific (between-species) variability, and that species can be characterised by mean trait values [4]. This has since been challenged by a number of studies [5, 6]. Trait-based research projects therefore need to critically consider whether to include intra-specific trait variability in their set-up, based on species, scale, and scope of the study [7].

The mechanisms that cause intra-specific variability as a species' response to its environmental conditions include adaptation via genetic changes and phenotypic plasticity [8]. Such mechanisms are expressed in a range of functional plant and seed traits [4, 9]. Functional traits have been defined as measurable features which interact with ecological factors through

under the grant number BU 3097/1-1. NC acknowledges past and current funding by Institut national de recherche pour l'agriculture, l'alimentation et l'environnement (INRAE,). The funders had no role in study design, data collection and analysis, decision to publish, or preparation of the manuscript.

**Competing interests:** The authors have declared that no competing interests exist.

specific functions in order to explain plant fitness components like growth, reproduction and survival [9, 10]. Sometimes, they are classified as hard and soft traits. A hard, usually physiological trait accurately describes a plant function, but is more difficult to quantify than a more easily measured, often morpho-anatomical soft trait. Soft traits can sometimes provide proxies for hard traits [11, 12].

Intra-specific variability arises at different scales and ecological organisational levels. It can be decomposed into parts [7]: the '*between-individual variability*' is the trait variability within a given population. The difference in trait values between populations can be called the '*population-level variability*' or '*between-population variability*'. The term 'population' refers in this to any group of sample individuals connected by a common environment: groups of plants growing in distinct locations (geographical regions, different habitats or even parts of a field), or groups of plants growing at the same location but stemming from different seed sources (e.g. a greenhouse or field experiment comparing different seed provenances).

The study of traits in weed research aims to understand processes which drive weed community composition and population dynamics. One of the main goals is to contribute to a more environmentally-friendly weed management [2, 13]. This includes attempts to make the modelling of weed populations generic rather than species-specific, based on a mechanistic representation of processes and involved species traits [3, 14, 15].

Arable fields, compared to other habitats of wild plant species, are characterised by a high frequency of disturbances through management and strong competition from crops. The right timing of germination and establishment, together with their ruderal growth and regeneration strategy are crucial for plant establishment and fitness in weed plants [16, 17].

Selection pressure favours species responses to local environmental cues that synchronise germination with optimal periods for seedling survival, establishment and reproduction [18–20]. Likewise, plants which compete better for light, nutrients and water have a higher fitness (i.e. the number of seeds produced per germinated seed), are more tolerant to resource deficits, or have higher plasticity in response to stresses [21].

The high level of disturbance and the ruderal strategy suggest a high adaptation capability in weed species, and consequently a high intra-specific trait variability. This must be accounted for in trait-based approaches in weed ecology to make them widely usable and valid in a more general context. Existing trait databases are extensive sources for trait information, but they are usually of limited use when it comes to intra-specific trait variation under variable environmental settings [1, 3].

Only a small number of studies investigated intra-specific trait variability in arable weeds, mainly for soft traits like canopy height, specific leaf area, or biomass. These studies concentrated on trait variability in response to different cropping systems, or explored the effect of variability on community assembly patterns and on ecosystem functioning [22–24].

Even fewer studies analysed intra-specific variability in traits associated to processes in population dynamics, like germination and early growth. In this paper, we aim to close this gap and analyse the intra-specific variability of two key traits in arable weed species: germination base temperature and early relative growth rate after emergence. Base temperature has been identified as one key parameter necessary for modelling weed seed germination [25], determining the days where germination is possible as well as controlling the speed of germination, pre- and post-emergence growth. The latter also depends on the early growth rate of seedlings which is an important ecophysiological trait in crop-weed-competition models [26].

Results of earlier studies were not consistent on the extent of intra-specific variability in these two traits, or other traits associated to germination [27–31]. Differences between populations were found for some species, not for others. Possibly, the population effect is only evident when populations come from locations with highly contrasted environments. No differences

were found between populations from near-by regions like Northern vs. Eastern France [32] and Central vs. Northern Italy [33, 34].

We therefore set up experiments to investigate, for a series of contrasting weed species, two seed populations originating from regions with contrasting environmental conditions. We expected to find considerable intra-specific trait variability in germination base temperature and seedling relative growth rate exist. We hypothesised that base temperatures as well as early relative growth rates would be lower in the seed populations from colder climate and lower solar radiation, due to adaptation to lower resource availability.

## Materials and methods

Two separate experiments were carried out in Rostock (Germany) between 2016 and 2017 (which are presented in detail here) following the protocols of earlier experiments carried out in Dijon (France) between 2009 and 2012 (already published [14, 21, 35]).

### Experiment on germination base temperature

Seed germination of 13 species was tested at four to six constant temperatures each (for details see S1 Table in S1 File). Seeds were laid out in Petri dishes (Ø 9 cm) lined with a double layer of filter paper and moistened with 5 ml deionised water or 10 mmol $KNO_3$ solution (*S. officinale*, *A. arvensis*) to break seed dormancy. Four replicates of 50 or 100 seeds per treatment were placed in temperature chambers with a 12-hour photoperiod. Petri dishes of *G. dissectum* were covered in two layers of aluminium foil as it requires to germinate in darkness, but could not be put in separate climate chambers without light cycle. Petri dishes were checked for germination at least once a day and moistened when the filter paper started to dry. Seeds were considered germinated once a radicle was clearly visible. Experiments lasted for approx. 4 weeks each, or until no more germination occurred during 7 days. Temperature in the chambers was monitored with data loggers (HOBO UX100-001/ Voltcraft PL-125-T2) every 10 minutes. The average of all measurements during the experiment was used for calculations.

Time-to-event models were fitted to the germination data [36]. We used a nonlinear and asymmetric three-parameter Weibull function following the equation

$$
\begin{aligned}
GP_t &= GP_{final} \times \left(1 - e^{-e^{b \times (\ln(t) - \ln(i))}}\right) \ for \ t > 0 \\
GP_t &= 0 \ for \ t = 0
\end{aligned}
\tag{1}
$$

where t is the time (in days) since water addition, $GP_t$ the proportion of seeds germinated up to the sampling time t, $GP_{final}$ the final proportion of seeds germinated out of the sample, i the point when inflection of the curve happens and b the shape parameter reflecting the steepness of the curve. We calculated $t_{50}$, which is the time to reach 50% of the final germination proportion from the fitted model parameters.

Germination rates (GR, in $d^{-1}$) were calculated as the inverse of $t_{50}$:

$$
GR = \frac{1}{t_{50}}.
\tag{2}
$$

A linear regression of the germination rate (in $d^{-1}$) over temperature (T in °C) was fitted per species:

$$
GR = a + b \times T.
\tag{3}
$$

The base temperature ($T_b$, in °C) was determined as the x-intercept of this regression line and calculated as

$$T_b = - \frac{a}{b} \tag{4}$$

with a and b being the intercept and slope of the regression [37].

### Experiment on relative growth rate in seedling stage

Early growth was monitored for 21 weed species in several experimental trials between spring 2016 and spring 2017 (Table 1), using the method developed by Colbach et al. [21]. Each trial included species that would germinate approximately around the same time in the field, i.e. spring germinating species in March/ April, early summer germinating species in May/ June and autumn germinating species in September/October. *V. arvensis* and *G. aparine* were tested in spring and in autumn. The experiment included seeds from populations in Northern Germany for all 21 species. For five summer-germinating species and one autumn-germinating species, seeds from French populations were tested, additionally.

Seeds were placed in Petri dishes, wetted with water or a $KNO_3$ solution and were set to germinate under optimal temperature and light conditions. When germination started, 20 seeds per species were placed in individual pots with standard garden earth, in a depth of 2–20 mm, depending on seed size, i.e. large seeds were sown deeper. Pots were placed in an unheated green house, with additional light (Philips sodium vapour lamps, 400 W) to maintain a

**Table 1. Species included in experiments on germination base temperature and early growth rate in Rostock 2016–2017.**

| Species | Experiment | | Season of growth rate experiment in Rostock |
|---|---|---|---|
| | Base temperature | Relative growth rate | |
| *Alopecurus myosuroides* | x | x | Autumn |
| *Amaranthus retroflexus* | - | x | Early summer |
| *Anchusa arvensis* | x | x | Spring |
| *Apera spica-venti* | x | x | Autumn |
| *Capsella bursa-pastoris* | - | x | Spring & Autumn |
| *Centaurea arvensis* | x | x | Spring |
| *Chenopodium album* | x | x | Early summer |
| *Digitaria sanguinalis* | - | x | Early summer |
| *Echinochloa crus-galli* | x | x | Early summer |
| *Galium aparine* | - | x | Autumn |
| *Geranium dissectum* | x | x | Spring |
| *Matricaria chamomilla* | x | x | Spring |
| *Matricaria inodora* | x | x | Spring |
| *Papaver rhoeas* | x | x | Spring |
| *Poa annua* | - | x | Autumn |
| *Setaria viridis* | x | x | Early summer |
| *Solanum nigrum* | - | x | Early summer |
| *Sonchus asper* | - | x | Autumn |
| *Stellaria media* | x | x | Spring |
| *Sisymbrium officinale* | x | - | Autumn |
| *Veronica hederifolia* | - | x | Spring |
| *Viola arvensis* | - | x | Spring & Autumn |

'x': species was tested, '-': species was not tested.

12-hour photoperiod. For six species (*A. retroflexus*, *Matricaria chamomilla*, *S. nigrum*, *S. media*, *V. hederifolia*, *V. arvensis*) the number of plants which initially survived in the pots was not sufficient for further analyses, therefore another 20 plants of each species were grown within two weeks after the start of the experimental run (and treated as a separate repetition). Temperature was recorded with data loggers (HOBO UX100-001/ Voltcraft PL-125-T2) every 15 minutes and averaged over the day. Pots were watered regularly, keeping the soil constantly moist to ensure unlimited growth.

Leaf area development was monitored by a non-destructive approach. One photograph was taken of each plant per day from above with a digital camera (Leica V-LUX 1, resolution 72 dpi). A red cardboard square of pre-defined area ($1cm^2$ or $4cm^2$) was placed beside the pot at soil height as a reference for scaling. Leaf area was then estimated from the green pixels on the picture with Easy Leaf Area software [38], scaling by the number of red pixels. Some of the 20 plants were harvested at 2-leaf, 4-leaf and 6-leaf stage respectively and their true leaf area determined using a flat bed desktop scanner (CanoScan LiDE 220). The ratio of leaf area on photograph to leaf area on scan was calculated per plant. A linear regression was fitted per species to this ratio over thermal time after emergence. Finally, the leaf area measurements from all photographs of the remaining plants were corrected using the estimated regression function.

The following equation was fitted to model leaf area development of each plant grown to the end of the experiment

$$LA_d = LA_0 \times e^{RGR \times TT_d} \tag{5}$$

where $LA_d$ is leaf area ($cm^2$) of the plant on sampling date d, $LA_0$ is (initial) leaf area at emergence ($cm^2$), $TT_d$ is thermal time from emergence to sampling day d (°Cdays, with a species-specific base temperature), and RGR the relative growth rate ($cm^2 \cdot cm^{-2} \cdot °Cdays^{-1}$). Leaf area was $log_n$-transformed and parameters fitted by linear regression over $TT_d$. Data points beyond the phase of exponential growth were left out of analysis. The end of this period was determined as the date when the local slope of logn(LA) vs $TT_d$ dropped below 1/10 of the slope at emergence (when $TT_d = 0$). Local slopes were calculated as the derivate of a polynomial fitted to logn(LA) vs $TT_d$ (S2 Fig in S1 File).

Thermal time $TT_d$ was calculated as

$$TT_d = \sum_{i=0}^{d} (T_i - T_b) \ for \ T_i \geq T_b \tag{6}$$

where $T_i$ is the mean temperature of day i (°C) and $T_b$ is the species-specific base temperature (°C). For twelve species, we used the germination base temperature obtained in our own experiment to calculate $TT_d$. For the other nine species we used literature values. Only plants for which the $R^2$ of the regression (Eq 5) exceeded 0.66 were used for further analyses to eliminate bad fits. For the calculation of averages, per-plant-values were weighted by the inverse of the relative standard error of each plant (i.e. se_LA0p/LA0p and se_RGRp/RGRp, with se_LA0p and se_RGRp the standard errors estimated from Eq 5).

## Species choice and seed material

For the experiments in Rostock we focused on the most frequent and important weeds in Northern German arable fields (Table 1). Seeds were mostly sourced from the experimental gardens and experimental fields of Rostock University, but in a few cases obtained from commercial suppliers aiming for seed lots from Northern Germany (S1 Table in S1 File). Seeds were stored in paper bags at room temperature and subjected to cold stratification prior to the experiments if necessary to break dormancy.

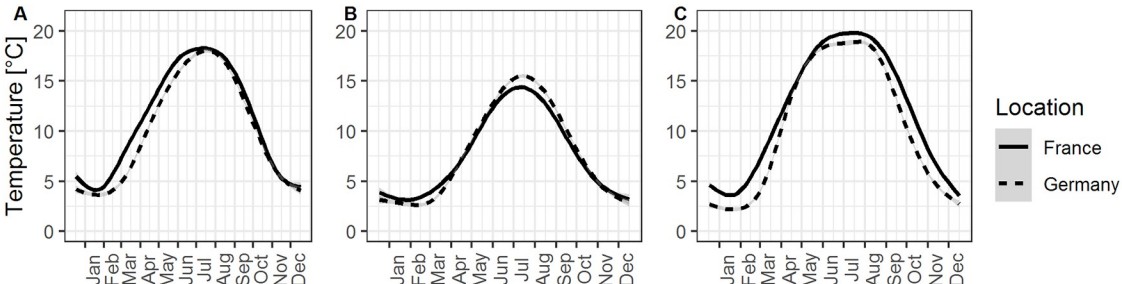

**Fig 1. Average daily temperatures in France (Bretenières near Dijon) and Germany (Warnemünde near Rostock) between 2000 and 2019.** A. Mean air temperature. B. Minimum air temperature. C. Mean soil temperature (10cm below ground). GAM smoothing over daily values, grey area is 95% confidence interval. Weather stations at the INRA experimental station of Dijon-Epoisses, and German Weather Service (DWD) Rostock-Warnemünde (Datasources: platform INRA CLIMATIK, www.opendata.dwd.de).

For six species, we tested seeds from the French populations (provided by the INRAE weed seed collection, UMR Agroécologie, INRAE Dijon) in Rostock, as an extension to the experiment. The results of these 'transplanted' seeds are presented in the S1 File.

**Conditions of seed-origin habitats.** Rostock/ Germany (54° 5′ N, 12° 8′ E, 14m asl) is located approximately 1200km northeast of Dijon/ France (47° 19′ N, 5° 3′ E, 257m asl). The climate in Rostock is characteristic for the transition zone between oceanic and continental climates with a mean temperature of 8.4°C and 591 mm precipitation. It has long winters with short periods of strong frost, a warm summer and a precipitation pattern with low amounts in spring and autumn and a maximum in July.

Dijon, in contrast has an oceanic climate with a continental tendency. Annual mean temperature is 10.5°C, with cold winters and hot summers. The precipitation of 732 mm is distributed nearly uniformly over all months, with a small maximum in May. Rostock has a shorter growing period, more days with a water deficit (evapotranspiration > precipitation) in spring and a higher probability of late frosts [39].

Mean air temperature is lower in Rostock than Dijon between January and October and similar at the end of the year (Fig 1). In contrast, soil temperatures are similar between mid May and August, but considerably lower in Rostock outside of summer.

## Analysis of inter-specific and intra-specific variability

**Data.** To analyse inter-specific and intra-specific variation of the two plant traits, we pooled the results from the experiments in Rostock with the experimental results obtained earlier in Dijon [14, 21]. The Dijon experiments were carried out on another, partly overlapping set of species, but we only use data for the species that were used in the experiments in Rostock.

The experiments were not initially planned and carried out with this kind of analysis in mind, i.e. rigorously including all possible factors into a randomly repeated experimental setup. We used advanced statistical methods to address the resulting causes of variability arising from not testing all species and populations at the same time and location, the hierarchical nature of the measurements in our datasets, as well as the unbalance in measurements in some of the hierarchical levels [40].

For clarity, we will refer to the locations of experiments as "Rostock" and "Dijon" in the further course of the manuscript, while "French" and "German" will refer to seed populations originating from the two different regions.

**Table 2. Sources of variation contributing to intra- and inter-specific trait variability included in the analysis.**

| Level of variability | | Trait experiment | |
| --- | --- | --- | --- |
| | | Base temperature | Early growth rate |
| Inter-specific | | • species | • species |
| Intra-specific | *between-population variability* | • seed population | • seed population |
| | | • experimental location | |
| | *between-individual variability* | | • experimental location |
| | | | • experimental conditions between repeated trials |

For explanations on 'between-population' and 'between-individual' variability see introduction.

We included four sources of variability in our analysis (Table 2). Note that base temperature as a trait is measured at population level, while early growth rate is measured for individual plants, hence the different assignment of "experimental location" in the categories.

**Inter-specific and intra-specific variability in germination base temperature.** The effects of species and population on germination base temperature were tested with a two-way analysis of variance (ANOVA). Pairwise tests were run on germination base temperatures between populations of the same species, comparing the estimates by calculating the z-score

$$z = \frac{T_{bG} - T_{bF}}{\sqrt{se\_T_{bG}^2 + se\_T_{bF}^2}} \tag{7}$$

where $T_{bG}$ and $T_{bF}$ are the germination base temperature estimates and $se\_T_{bG}$ or $se\_T_{bF}$ the corresponding standard errors for German and French populations, respectively. The significance p of the difference was calculated as

$$p = 1 - 2 \int_{u=-\infty}^{z} \frac{1}{\sqrt{2\pi}} e^{\frac{-u^2}{2}} du. \tag{8}$$

**Inter-specific and intra-specific variability in relative growth rate.** Mixed effect models were used to analyse early relative growth rates with species, seed population and experimental location as fixed factors and repetition as a random factor to account for the nested structure of the data and the unbalanced number of plants (3 to 10) within each combination of species, population, location, and repetition [40]. A Type III ANOVA with Satterthwaite's method for denominator degrees of freedom and F-statistic was used to test the significance of effects of species, population and experimental location.

We also compared intra-specific and inter-specific variability using variation coefficients. First, we averaged measured relative growth rates per species and population. Then we calculated the coefficient of variation between population means per species (intra-specific variation) or between species means per population (inter-specific variation).

We estimated marginal means to specify the relative growth rate per species and population and conducted a pair-wise comparison of means to test for differences between populations. If the difference was not significant, we finally estimated the species relative growth rate from both populations combined.

Within the paper, we present analyses of relative growth rate variability without the data collected from French seeds in the Rostock experiment. The results differed only slightly when incorporating these "transplanted seeds" and are included in the S1 File. We decided against using this data in the main analysis because it made our dataset even more unbalanced (by

growing seeds in an environment different to their origin and only transplanting one population, not both).

**Relationship between base temperature and relative growth rate.** We tested the relation between base temperature, seed population and early relative growth rate (RGR) by fitting a linear model for each population, followed by an ANOVA.

**Software.** All statistical analyses were carried out with R 3.1.6 [41], germination curves fitted with package *drc* [42], linear mixed models fitted with packages *lme4* [43] and *lmerTest* [44], and marginal means estimated using package *emmeans* [45].

## Results

### Germination base temperatures

All regressions correlating germination rates to temperatures had good fits, with $R^2$ ranging from 0.81 to 0.99 (Table 3). Base temperatures varied between -0.2˚C (*Apera spica-venti*) and 11.9˚C (*Setaria viridis*).

For the six species for which we had measurements of both seed populations, the base temperatures for the German populations were all clearly higher than for the French populations (Fig 2), with a significant difference for two species (Table 3). In an analysis of variance for the same data, species and population both had significant effects on germination base

**Table 3. Germination base temperature of 13 weed species from German seed populations.**

| Species | Main emergence periods in fields | Base temperature (˚C)– German population | $R^2$ | Base temperature (˚C)– French population | p-value of difference | Ranges of base temperature from other sources (˚C) |
|---|---|---|---|---|---|---|
| *Alopecurus myosuroides* | G: autumn | 1.9 ± 0.7 | 0.95 | 0.0 [32] | 0.31 | - |
| | F: autumn & spring | | | | | |
| *Amaranthus retroflexus* | early summer | 10.5 ± 1.3 | 0.86 | 8.9 ± 1.1 [35] | 0.35 | 10–12.6 [34, 46] |
| *Anchusa arvensis* | G: spring & autumn | 8.3 ± 2.0 | 0.81 | nt | | |
| *Apera spica-venti* | G: Early spring & autumn | -0.2 ± 0.9 | 0.92 | nt | | |
| *Centaurea arvensis* | Autumn | 3.3 ± 1.0 | 0.81 | 2.2 ± 1.2 [47] | 0.48 | 1.1–1.9 [29] |
| *Echinochloa crus-galli* | early summer | 10.0 ± 0.6 | 0.92 | 6.2 ± 0.6 [35] | **< 0.001** | 10.4–13.2 [34, 48, 49] |
| *Geranium dissectum* | G: spring & autumn | 4.0 ± 0.4 | 0.93 | 0 ± 1.3 [35] | **0.003** | - |
| | F: autumn | | | | | |
| *Matricaria chamomilla* | G: spring & autumn | 4.7 ± 0.6 | 0.96 | nt | | - |
| *Matricaria inodora* | spring & autumn | 2.9 ± 0.5 | 0.96 | 2.0 ± 0.6 [35] | 0.25 | - |
| *Papaver rhoeas* | G: spring & autumn | 2.5 ± 0.2 | 0.99 | nt | | 1.0 Spain [50] |
| *Setaria viridis* | early summer | 11.9 ± 0.5 | 0.96 | nt | | 6.1–12.5 [48, 51] |
| *Sisymbrium officinale* | G: autumn | 5.7 ± 0.2 | 0.96 | nt | | |
| *Stellaria media* | G: all year | 2.3 ± 0.2 | 0.98 | nt | | 1.4 [52] |
| | F: spring & autumn | | | | | |

Values are estimates +/- standard error. Comparison to values from French populations and/or literature. Significant differences between German and French populations in bold type. Main emergence periods given for Northern Germany (G) and Eastern France (F). Periods are similar for both provenances if not marked otherwise.

nt = not tested.

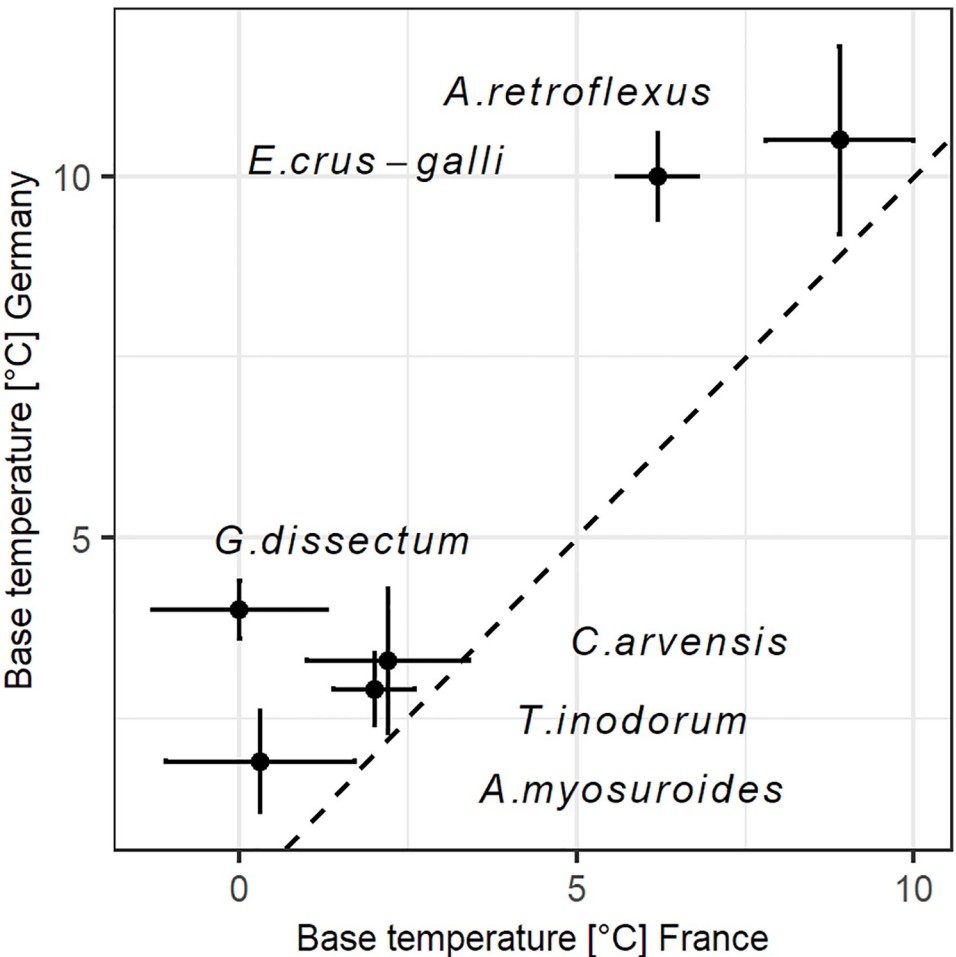

**Fig 2. Germination base temperatures of six weed species comparing two populations from contrasting provenances.** The dashed line marks the ratio 1:1. French populations tested in Dijon, German populations in Rostock, using growth chambers with similar conditions in both cases.

temperature (partial $R^2$ = 0.88, p = 0.001 and partial $R^2$ = 0.10, p = 0.012 respectively). The variability in base temperature was higher among species than between populations.

## Relative growth rates

Relative growth rates were strongly dependent on species which accounted for about a third of the variation ($\eta^2$ = 0.303, p<0.001). Differences in the experimental conditions between (repeated) trials explained nearly half the variance in the data (Intra-correlation Coefficient ICC = 0.47), but experimental location had no significant effect (S2 Table in S1 File). Variation in relative growth rate between species was about three times the variation within species (Fig 3).

Seed population had no overall effect in our model (p = 0.93). We found a significant species:population interaction ($\eta^2$ = 0.151, p <0.001) accounting for half as much variation as species. The pairwise comparison of means between French and German populations of each species showed a significant difference for one species, *S. nigrum* (S4 Table in S1 File). In around half of the species we found a higher growth rate for the German seed population, in

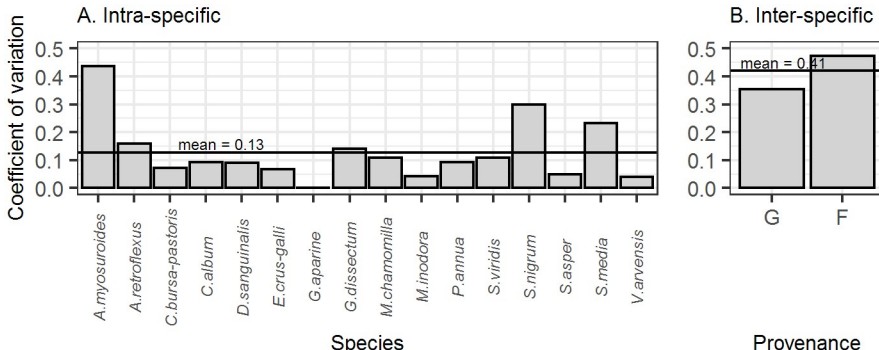

**Fig 3. Components of variability in relative growth rate of 16 arable weed species.** A. Intra-specific variability: coefficient of variation among population averages of all plants from the same species. B. Inter-specific variability: coefficient of variation between species averages of all plants from the same seed population. One population per species and provenance: G: German, F: French.

the other half it was higher for the French population. Estimates of the species-specific relative growth rate varied between 0.0101 (*V. hederifolia*) and 0.0446 cm$^2$cm$^{-2}$°C$^{-1}$d$^{-1}$ (*S. nigrum*) (S8 Table in S1 File).

## Relationship between base temperature and growth rate

We found a positive relationship between germination base temperature and early relative growth rate in the studied species (Fig 4). Species, which germinate at higher temperature, grow more per degree-day than species with low germination base temperature. A linear model for the French populations had a fit of $R^2 = 0.74$ and for German populations of $R^2 = 0.85$. The slopes were not significantly different when testing for the influence of seed population on the relation between the two variables (t = -2.022, p > 0.05).

## Discussion

We reported on the outcome of two germination and early growth experiments, comparing contrasting species as well as seed populations of given species originating from contrasting climate conditions.

### Germination base temperatures are higher in colder-climate populations

Our results on base temperature of weed species from Northern German provenance are consistent with the ranges reported for European arable weed species in the literature (Table 3). We found that typical summer-germinating weed species (which originate from warmer regions) presented a considerably higher base temperature than the spring and autumn germinating species. At the inter-specific level, our findings are thus consistent with the hypothesis of higher environmental temperatures leading to higher germination temperature requirements which has been confirmed for many organismal groups [53].

Unexpectedly, at the intraspecific level (i.e. for a given species), we found that base temperatures were higher for seeds originating from the colder climate (i.e. Rostock) than from the warmer Dijon. A negative relationship between ambient temperature and germination temperature requirements of populations of the same species, though, has been found by other authors as well, both for a North-South gradient [54] and an altitudinal gradient [55]. The germination base temperature seems to be adapted to prevent seeds from germinating in unfavourably cold conditions.

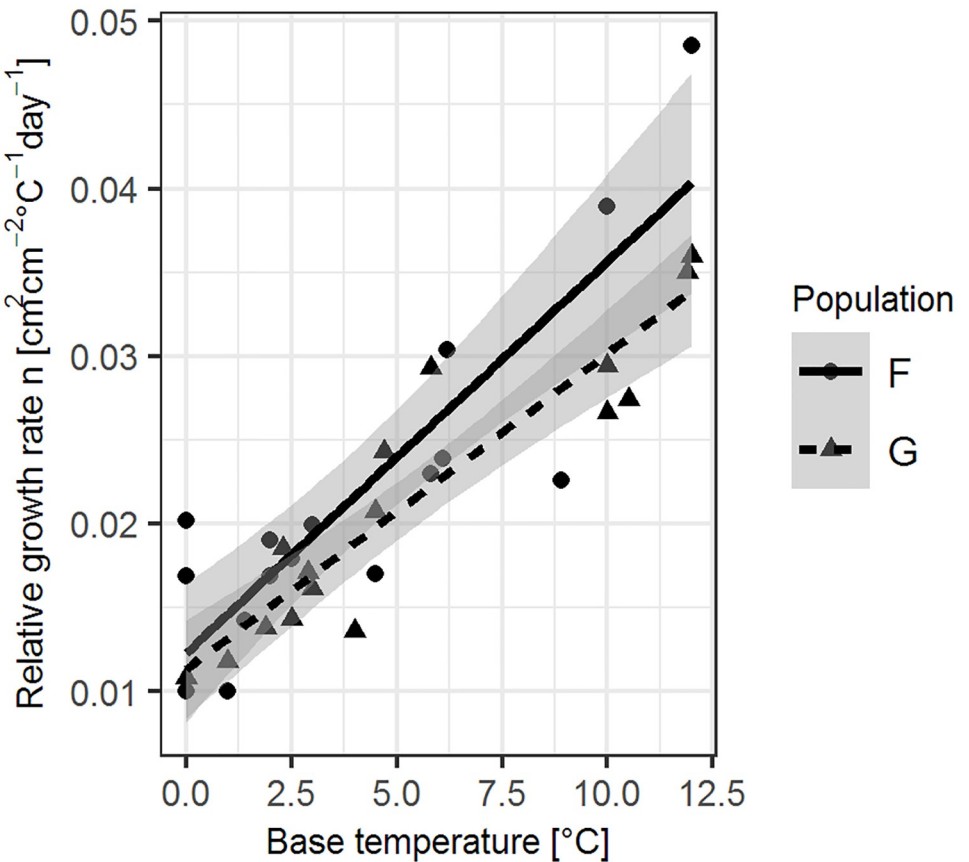

**Fig 4. Relationship between germination base temperature and relative growth rate per species and population.** Fitted functions: German seeds: RGR = 0.011 + 0.0019 x $T_b$, $R^2$ = 0.86; French seeds: RGR = 0.012 + 0.0023 x $T_b$, $R^2$ = 0.74.

Possibly, our results were also influenced by the environmental conditions at seed set and during seed development of our samples [54, 56]. Such maternal effects on seed germination behaviour have been reported, e.g. faster and/or more germination in case of nitrogen deficiency and water stress before or during seed set [57, 58]. However, it would be necessary to use a common garden approach in order to distinguish between maternal effects vs. genetic adaptation [30, 33, 59].

### Base temperature may control germination only for spring emergence periods

Germination base temperature has been generally assumed as giving a cue of favourable growing conditions for seeds to start germination and emergence, but also halt germination in autumn before the seedlings' ability to survive winter could be decreased by late germination [59–61]. As in a previous Burgundy study [14], our experiments suggest that this relationship may only hold true for the spring emergence periods and therefore for species that are facultative or strict spring-germinators. Indeed, emergence generally starts earlier in spring and ends later in autumn in Dijon than in Rostock (Fig 5) and, likewise, in Sweden [29]. In more Northern locations, species need to avoid cold and frost injury to their sensitive seedlings (thus explaining spring germination at higher temperatures) whereas in more Southern habitats

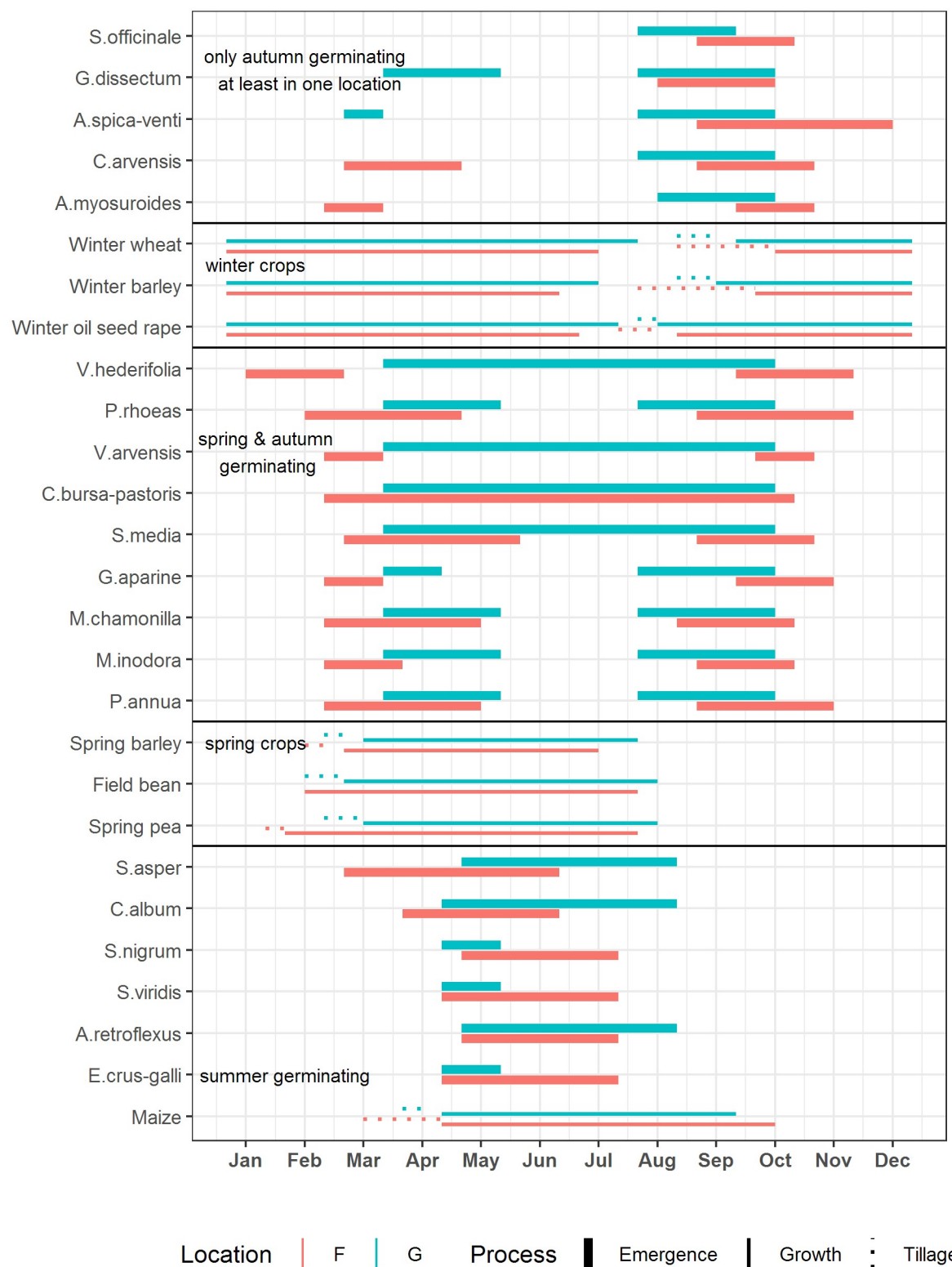

**Fig 5. Main emergence periods of common arable weed species compared to field times of typical crops in North Eastern Germany and Burgundy.** (Expert knowledge [14], Pulkenat pers. comm.).

drought and hot temperatures present the highest risk for seedling establishment [62] (explaining the later start of fall emergence).

Of the two functions of base temperature, triggering germination in spring part may be more important for population fitness than shutting off germination in late autumn. Autumn emergence usually starts as soon as the soil is moist enough, and when temperature decreases down to the base temperature (irrespective of its value) later in the season, most non-dormant seeds have already germinated and grown to a stage tolerant to frost. In spring on the other hand, a low base temperature would let too many seeds germinate at a time when temperature may still quickly fall below zero, potentially leading to high losses due to frost.

We find more evidence for this mechanism in species whose germination habit differs between sites. Such shifts have been reported for a number of species [61]. *G. dissectum* germinates only in autumn in Dijon, but in Rostock also in spring (Fig 5), which may explain the large difference of base temperature for *G. dissectum* that we found (4˚C in Rostock vs. 0˚C in Dijon) [63]. We suspect *G. dissectum* to be (mostly) dormant in spring in Dijon. The base temperature therefore has no function to prevent seeds from germinating too early in spring, and subsequently selection pressure may never have favoured adaptation towards a higher base temperature. Similar differences in germination timing between provenances have been reported for *C. bursa-pastoris* [64] and *C. canadensis* [59].

## Germination in early and late summer is controlled by the interaction of climate, weather and crop management

The emergence timing of weeds in early and late summer depends on crop management and its interaction with weather. The climatic gradient between the seed sources in our study also causes differences between the agricultural management systems in those regions.

Indeed, summer germinating weed species are the only ones that emerge at the same time in Rostock and Dijon, despite higher base temperatures for German seed populations. But, summer germinators are also the only species whose favourite crop (i.e., maize) is sown synchronously at the two locations (Fig 5). Likewise, disparities between location-specific emergence patterns can arise due to different soil and crop management practices [65] and the pattern eventually is memorized by genetic adaptation. For instance, the later start of tillage and cultivation for winter-sown crops in Dijon (optimizing crop management) thus selects within weed species for populations that emerge later (after last tillage). A similar effect of management timing on germination was found for *Datura stramonium* seeds from three Southern European populations [33].

## Intra-specific variability in early growth rates is much smaller than inter-specific variability

Early relative growth rates for three of the four species measured in Rostock agree with literature values [21]. Only the growth rate for *V. hederifolia* was slower than in any other previously measured weed species. Other studies on early growth rate often measured above-ground biomass rather than leaf area [21, 26, 66–70], which makes it hard to compare ranges.

To our knowledge, this study is the first to investigate inter-population differences in early growth rates of (non-woody) plant species, apart from [31]. Contrary to the previous results however, inter-specific variability in relative growth rates was here much larger than intra-specific variability. This is partially related to the original species choice in Dijon which aimed to investigate species contrasting in various characteristics like emergence time, seed mass, guild (grass or herb). In any case, we found a significant population effect on relative growth rate

only for one species. This result is in agreement with earlier studies which found no maternal effects on life cycle stages later than germination [56].

In general, the inter-population or intra-specific variability decreased when a species was tested more than once, probably by a combination of lowering the experimental error as well as exploring a wider range of growing conditions with each population leading to more overlap. However, for some species, we found a very high variability in growth rates between successive trial with the same species and population indicating a plastic response to varying environmental conditions [26].

Our experiments only investigated early post-emergent growth in unlimited resource conditions. In contrast to our results, studies focusing on later growth and/or stressful conditions reported more intra- than inter-specific variability, whether in terms of seed provenance and experimental location [71] or plasticity in response to stress [21]. Moreover, we only worked with two locations but 15 species (chosen for their contrasting behaviour) (vs. 4 species, but several locations in Europe in [71]). It is therefore important when testing intra-specific variation to consider choosing populations from the whole (geographic) distribution of a species.

## Early relative growth rate measured in relation to thermal time may ignore sensitivity to light quality

For *A. myosuroides*, the high intra-specific variability in relative growth rate (RGR) (Fig 3) might be caused by the differences in light availability during the experimental runs. These were carried out in the two locations in different seasons of the year (autumn in Rostock, spring in Dijon) to account for the differences in species emergence season (strict autumn germinator in Germany, facultative autumn and spring germinator in Burgundy). Indeed, seedlings of the same species grown in autumn were reported to have a lower RGR per degree-day than the ones that emerged in spring which was explained by the lower light availability [26]. This also explains why we found that plants in consecutive spring/summer months had higher RGRs (S4 Fig in S1 File).

The benefit from increased light availability for plants growing in longer days with higher solar radiation [70] also contributes to explaining the positive relationship between base temperature and relative growth rate found here and in another analysis of 49 species tested in Dijon [21]. Indeed, a higher base temperature delays emergence to longer and sunnier days. The other explanation is the trade-off between base temperature and rate of biological processes reported also for other organisms [53].

## Conclusion

We have shown that local adaptation can be strongly trait-specific, raising the question whether certain traits are under higher selection pressure or are simply more plastic. Regarding implications for future weed models, our results provide additional criteria for explicitly incorporating intra-specific trait variability within the part simulating germination. In the context of typical weed modelling research where the scale of study is often local to regional and centred on sites with a weed community rather than one single species, intra-specific trait variability should be included if the focus is on response traits like the community assembly. It might be negligible if the focus is laid more on effect traits such as ecosystem functioning or net primary productivity.

We suggest to increase accuracy of future modelling exercises within a study region by using germination base temperatures from local seed populations. There seems to be no need to specifically measure early growth rates from local populations, although further experiments with new populations increase the precision of species trait averages and ranges.

## Supporting information

**S1 File.**
(DOCX)

## Author Contributions

**Conceptualization:** Jana Bürger.

**Data curation:** Jana Bürger, Nathalie Colbach.

**Formal analysis:** Jana Bürger.

**Funding acquisition:** Jana Bürger.

**Investigation:** Jana Bürger.

**Methodology:** Nathalie Colbach.

**Project administration:** Jana Bürger.

**Visualization:** Jana Bürger.

**Writing – original draft:** Jana Bürger.

**Writing – review & editing:** Jana Bürger, Andrey V. Malyshev, Nathalie Colbach.

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
