## [Decision Letter · Decision Letter 0]

14 Jul 2020

PONE-D-20-16133

Arable weed species show local adaptation in germination base temperature but not in seedling growth rate

PLOS ONE

Dear Dr. Bürger,

Thank you for submitting your manuscript to PLOS ONE. After careful consideration, we feel that it has merit but does not fully meet PLOS ONE’s publication criteria as it currently stands. Therefore, we invite you to submit a revised version of the manuscript that addresses the points raised during the review process.

ACADEMIC EDITOR:

I have now had time to evaluate the reports submitted by 2 referees on your submitted manuscript. Both the referees think that your work is worth publishing; however, suggested MAJOR revisions.For example, both referees think that the data of France can be excluded from the manuscript for easier readability of the manuscript. Nonetheless, the current version contradicts with the hypothesis.I suggest to use same populations (at least include in the manuscript) for both traits studied in the manuscript.Revise your manuscript carefully in the light of the comments. I look forward to receive your revised manuscript.

We look forward to receiving your revised manuscript.

Kind regards,

Shahid Farooq, Ph.D.

Academic Editor

PLOS ONE

Journal Requirements:

2. Please ensure that you refer to Figure 4 in your text as, if accepted, production will need this reference to link the reader to the figure.

3. Please include a copy of Table 5 which you refer to in your text on line 288.

Reviewers' comments:

Reviewer's Responses to Questions

**Comments to the Author**

1. Is the manuscript technically sound, and do the data support the conclusions?

Reviewer #1: Yes

Reviewer #2: Yes

2. Has the statistical analysis been performed appropriately and rigorously? 

Reviewer #1: Yes

Reviewer #2: No

3. Have the authors made all data underlying the findings in their manuscript fully available?

Reviewer #1: Yes

Reviewer #2: Yes

4. Is the manuscript presented in an intelligible fashion and written in standard English?

Reviewer #1: Yes

Reviewer #2: Yes

5. Review Comments to the Author

Reviewer #1: Generally, a good research has been done to provide useful information on the germination of some important weeds, especially in Germany. There were some problems with the material and the method, which I wrote in the main file of manuscript. My most important suggestion is to exclude the experiment in France from manuscript and only present the results of the experiment in Germany, which is due to the following two cases. First of all, the complexity of the article is lessened and it is easier for the reader to understand it. Secondly, the repetition has not been done properly. Therefore, after deleting the results of the French experiments and re-analyzing the data, manuscript should be resubmitted That's why I didn't check the results section because it really causes headaches.

Reviewer #2: I have evaluated the manuscript PONE-D-20-16133 submitted by Bürger et al. Overall it is an interesting study which would contribute towards the development of Weed Ecology. There are several major flaws which need to be addressed before the manuscript can be published. I am interested to review the revised version of the manuscript.

1. I suggest to exclude the results for French side due to 2 reasons. First, it seems that you are doing inter-specific variaiton studies and secondly it makes the manuscript complex. Keep it simple and include only results from Germany.

2. There are different number of species in both studied traits. Is it possible to balance and further possible to include same species for both traits?

3. There is no info in MM which constant temperatures were used.

4. Results and discussion are too long. Please reduce them by 50%

5. Restate conclusion based on the revised statistical analysis.

Please see attached file for other comments.

6. PLOS authors have the option to publish the peer review history of their article (what does this mean?). If published, this will include your full peer review and any attached files.

Reviewer #1: No

Reviewer #2: No

---

## [Author Response · Author response to Decision Letter 0]

24 Sep 2020

Dear Shahid Farooq,

we have now had time to revise our manuscript according to your recommendations as well as the reviewers’ comments. We have changed the title, taken strong efforts to increase the readability, and edited minor errors.

We would like to answer to the main comments within this letter. Other comments are answered in the annotated versions of the reviewer’s pdfs.

1.) Editor’s Recommendations:

For example, both referees think that the data of France can be excluded from the manuscript for easier readability of the manuscript.

-> Omitting the data from France would mean to omit the intra-specific trait variability between seed populations. As the aim of the analysis is explicitly to look at this kind of variability it does not seem practical to just use German data.

The question of readability arose due to the length of the manuscript, particularly results section. Could you please condense the results keeping the results of both regions, i.e., Germany and France for improving the readability for a layman? I guess this will solve the issue and comment of both reviewers regarding exclusion of France from the analysis.

-> The results section was shortened, for example by moving Table 5 to the Supporting Material.

Nonetheless, the current version contradicts with the hypothesis. Regarding the hypothesis issue, the possible solution could be to change the title a bit, which can be reflective of both intra and inter-population variation.

-> Our results indeed contradicted the hypothesis, which should happen fairly regularly in research. We chose a new title to the manuscript and included “intra-specific variability”. 

I suggest to use same populations (at least include in the manuscript) for both traits studied in the manuscript.

-> Do you suggest to use the same sets of species for analysis of the two experiments? Unfortunately, this would diminish the data base for the analyses a lot. Both experiments were quite different in their resource requirements (room, time, timing, seed numbers). The base temperature experiment is also more prone to errors and failures than the growth rate experiment. Therefore, we could handle a much lower number of species in the base temperature experiment than in the growth rate experiment.

-> Although the larger set of species in the growth rate analysis may be irritating at first, it increases the power of the analysis: even though we have 21 contrasting species in the analysis, we find no general evidence for local adaptation in this trait. On the other hand, local adaptation in base temperature is evident even when we only look at 6 species.

Please ensure that you refer to Figure 4 in your text as, if accepted, production will need this reference to link the reader to the figure.

-> Reference was included.

Please include a copy of Table 5 which you refer to in your text on line 288.

-> Table 5 was moved to supporting Material, and reference changed accordingly.

2.) Reviewer # 1

There were some problems with the material and the method, which I wrote in the main file of manuscript.

-> see comments there.

My most important suggestion is to exclude the experiment in France from manuscript and only present the results of the experiment in Germany, which is due to the following two cases. First of all, the complexity of the article is lessened and it is easier for the reader to understand it. 

-> please see comment to Editor regarding exclusion of French data.

-> We thoroughly revised the manuscript, mainly in the Materials and Methods section, subsection on statistical analysis to clarify and make reading easier.

Secondly, the repetition has not been done properly. 

-> A paragraph was included explaining that our study is not based on a repeated experimental design, but rather uses advanced statistical methods to account for any problems of combining data from different sources.

Therefore, after deleting the results of the French experiments and re-analyzing the data, manuscript should be resubmitted That's why I didn't check the results section because it really causes headaches.

3.) Reviewer #2

I have evaluated the manuscript PONE-D-20-16133 submitted by Bürger et al. Overall it is an interesting study which would contribute towards the development of Weed Ecology. There are several major flaws which need to be addressed before the manuscript can be published. I am interested to review the revised version of the manuscript.

1. I suggest to exclude the results for French side due to 2 reasons. First, it seems that you are doing inter-specific variaiton studies 

-> Our analysis contains both inter-specific and intra-specific variability. Especially for the growth rate experiment, this gives more power to our analysis: on one hand we can show that there is little intra-specific difference between French and German populations of the same species; on the other hand, we can show that we used a wide set of contrasting species. Finally, we could show that the inter-specific variation even within each region is three times higher than the intra-specific variation for any of the single species.

and secondly it makes the manuscript complex. Keep it simple and include only results from Germany.

-> please see comment to Editor regarding exclusion of French data.

2. There are different number of species in both studied traits. Is it possible to balance and further possible to include same species for both traits?

-> please see comment to editor for the same suggestion

3. There is no info in MM which constant temperatures were used.

-> This information was included in the Supporting Material because it is a lot of detail. We added a reference to the appropriate table within the M&M text.

 4. Results and discussion are too long.

-> We revised the whole manuscript, aiming at a concise language and discussion, but with necessary detail regarding that experiments and analysis are complex. The discussion was shortened by 15%, some tables transferred to the supporting information. Overall, the manuscript now has less than 8000 words (including all references).

5. Restate conclusion based on the revised statistical analysis.

-> The intended analysis is not possible without the data from France. We therefore would like to keep the French data as well as the conclusion.

Lastly, we were asked by the editorial office to explain the abbreviation INRAE, the institution of our

co-Author Nathalie Colbach. INRAE is short for Institut national de recherche pour l’agriculture,

l’alimentation et l’environnement.

We hope we could answer to all problems and hope to receive a positive vote to publish our article in its revised form.

Thank you very much.

On behalf of all authors,

Jana Bürger

---

## [Editor Report · Decision Letter 1]

29 Sep 2020

Populations of arable weed species show intra-specific variability in germination base temperature but not in early growth rate

PONE-D-20-16133R1

Dear Dr. Bürger,

We’re pleased to inform you that your manuscript has been judged scientifically suitable for publication and will be formally accepted for publication once it meets all outstanding technical requirements.

Kind regards,

Shahid Farooq, Ph.D.

Academic Editor

PLOS ONE

Additional Editor Comments (optional):

I have now gone through the revised manuscript. I commend authors for thoroughly revising their manuscript and addressing all comments. The current version can be accepted for publication in Plos One.
---

## [Editor Report · Acceptance letter]

1 Oct 2020

PONE-D-20-16133R1 

Populations of arable weed species show intra-specific variability in germination base temperature but not in early growth rate 

Dear Dr. Bürger:

I'm pleased to inform you that your manuscript has been deemed suitable for publication in PLOS ONE. Congratulations! Your manuscript is now with our production department. 

Kind regards, 

on behalf of

Dr. Shahid Farooq 

Academic Editor

PLOS ONE